



# Comparing float $p$CO$_2$ profiles in the Southern Ocean to ship data reveals discrepancies

Chuqing Zhang[1], Yingxu Wu[2], Peter J. Brown[3], David Stappard[1], Amavi N. Silva[1], and Toby Tyrrell[1]

[1]University of Southampton, National Oceanography Centre Southampton, Southampton, UK
[2]Polar and Marine Research Institute, Jimei University, Xiamen, China
[3]National Oceanography Centre, Southampton, UK

*Correspondence to*: Chuqing Zhang (Chuqing.Zhang@soton.ac.uk)

**Abstract.** The Southern Ocean plays a crucial role in the global carbon cycle. Recently, the utilization of biogeochemical
(BGC) Argo float data has provided valuable insights into the uptake and release of carbon dioxide (CO$_2$) by this region.
However, significant uncertainty remains regarding the accuracy of $p$CO$_2$ (partial pressure of CO$_2$) values derived from float
data. In this study, we compared $p$CO$_2$ estimates obtained from float pH data with those from ship-collected data across the
Southern Ocean, employing $p$CO$_2$-depth, $p$CO$_2$-O$_2$ and CO$_2$-O$_2$ vs saturation plots to assess the degree of agreement between
these two datasets. Our findings reveal significant systematic differences. A preliminary analysis, ignoring other factors,
found that the float data is consistently higher, on average, than the ship data at equivalent depths and oxygen levels. We
tested the hypothesis that inaccurate float pH data or float $p$CO$_2$ correction process is the main cause of the $p$CO$_2$ difference,
by quantifying other factors that could produce systematic differences, including: (i) spatial sampling bias, (ii) seasonal bias,
(iii) errors in estimated alkalinity, (iv) errors in carbonate system constants, and (v) higher levels of anthropogenic CO$_2$ in
float data. However, none of the other factors were found to be able to fully account for the discrepancies, suggesting issues
with float pH data quality and/or the float $p$CO$_2$ correction process. Additional analysis included refinements to ship-based
and float-based $p$CO$_2$ before intercomparison. Overall, we estimate that, in the Southern Ocean, surface $p$CO$_2$ from floats is
biased high by, on average, at least 10 µatm.

## 1 Introduction

The Southern Ocean (SO) is a substantial carbon sink, capable of absorbing and storing globally-significant quantities of
carbon dioxide (Rintoul, 2011), a service critical to the mitigation of global climate change. The Southern Ocean
disproportionally accounts for roughly 40% of the global oceanic uptake of atmospheric CO$_2$ while accounting for only 25%
of its surface area (Landschützer et al., 2015, Gruber et al., 2019). Meanwhile, upwelling in the SO also leads to the release
of carbon dioxide to the atmosphere (Rae et al., 2018). Accurate calculation of CO$_2$ fluxes has profound implications for
analyzing the carbon cycle and understanding carbon dynamics (Bauer et al., 2013). Typically, the flux is determined by
measuring the difference in CO$_2$ partial pressure ($p$CO$_2$) between surface waters and the overlaying air. Historically,





previous calculations of surface $CO_2$ fluxes were based on high-quality shipboard data (McNeil et al., 2007, Metzl et al., 2006, Lenton et al., 2013) but ocean measurement equipment developed in recent years has opened up new avenues for the determination of $p$$CO_2$ and $CO_2$ fluxes, including biogeochemical (BGC) Argo floats, aircraft, and uncrewed surface vehicles (Claustre et al., 2020, Long et al., 2021, Sutton et al., 2021). BGC floats hold great potential as an efficient tool for

carbon cycle research particularly in the inhospitable SO, as they can rapidly generate a substantial volume of data with broad spatial and temporal coverage, bridging gaps in the sparse shipboard observations (Johnson et al., 2017).

Up to December 2022, 493 BGC Argo floats had been deployed across the global oceans, ([https://www.go-bgc.org/array-status#current-deployments](https://www.go-bgc.org/array-status#current-deployments)), equipped with biological and chemical sensors to measure a combination of oxygen ($O_2$), nitrate, pH and other variables over a lifespan of several years (Owens et al., 2022). Measurements (vertical profiles) are

made every 5-10 days and the resulting data is useful for investigation of marine biogeochemical processes, including ocean acidification, carbon cycling, and air-sea gas exchange (Matsumoto et al., 2022). As $p$$CO_2$ is not currently measured directly, it is estimated using a combination of float-measured pH and algorithm-estimated total alkalinity (TA) (Bushinsky et al., 2019, Williams et al., 2017). When float-based estimates of $p$$CO_2$ are incorporated into large-scale estimates of carbon flux in SO, substantial disparities in the magnitude of the flux have been revealed compared to those estimates derived solely

from ship-based measurements or from non-marine data (Long et al., 2021, Gray et al., 2018, Bushinsky et al., 2019). Using solely BGC float data, an annual $CO_2$ flux of -0.08 Pg C/yr was calculated by Gray et al. (2018), while combining with ship data led to an annual mean Southern Ocean (south of 35°S) sink of $-0.35 \pm 0.19$ Pg C/yr being calculated for the years 2015–2017 (Bushinsky et al., 2019). These estimates compare to a value of $-1.14 \pm 0.19$ Pg C/yr based entirely on ship data (where negative values indicate net carbon fluxes into the ocean), meaning the annual net flux based entirely on float data is

considerably smaller than the value based on ship data.

What leads to this difference between float-based and ship-based $CO_2$ flux estimates? Bushinsky et al. (2019) attributed it to ships lacking data from winter observations (when $p$$CO_2$ values are the highest) at high Antarctic latitudes. However, Mackay and Watson (2021) found their estimates of winter outgassing (extrapolated from summertime ship observations) to be much smaller than the float-derived observations and close to long-term ship data estimation results. Moreover, Long et al.

(2021) calculated the ocean-atmosphere $CO_2$ flux using data collected from aircraft and found it to be close to the ship-based $CO_2$ flux but to significantly different from the float-based $CO_2$ flux (aircraft-based annual mean flux for 2009–2018 of -0.53 $\pm$ 0.23 Pg C/yr, compared to float annual mean of around +0.3 Pg C/yr for 2015–2017) in south of 45°S. Furthermore, Wu and Qi (2023) highlight the no seasonal difference throughout the year feature of $CO_2$ flux difference between float-based and ship-based, which could not be explained by the hypothesis Bushinsky et al. (2019) proposed. These results raise the

possibility that the difference in $CO_2$ fluxes may stem from the calculation of $p$$CO_2$ from float pH data.

A fundamental challenge associated with BGC float pH data is the limited opportunity for maintenance and servicing once deployed in the ocean, despite potential exposure to various contaminants and risks of damage during their extended lifetime of oceanic measurements. Factors such as bio-detritus deposition can introduce biases into the collected float data (Bittig et al., 2019, Claustre et al., 2020). In contrast, equipment used for acquiring ship-based data undergoes frequent and repeated



maintenance, field testing, and calibration and is thus generally considered to be more reliable than float-derived data (Pierrot et al., 2009, Williams et al., 2017).

Williams et al. (2017) identified a large number of potential sources of float $pCO_2$ uncertainty, which they grouped into 3 categories: (1) uncertainties in pH sensors, (2) uncertainties in derived alkalinity estimates, and (3) uncertainties in carbonate system equilibrium constants, with (1) having the greatest contribution. Although there is a quality control and calibration

process applied to float pH data, including linear drift correction and crossover comparison (with co-located ship data) correction (Maurer et al., 2021), questions still remain regarding the calibration process and its outputs (Wu et al., 2022, Álvarez et al., 2020, Huang et al., 2023). Álvarez et al. (2020) highlighted the mismatch between float-based and ship-based pH values. Mackay and Watson (2021) reconstructed winter $pCO_2$ data based on ship-collected summer data, and found them to diverge somewhat from the float pH data based $pCO_2$. Sutton et al. (2021) concluded that float $pCO_2$ data exhibit

greater uncertainty than ship and uncrewed surface vehicle-collected $pCO_2$ data. Moreover, Wu et al. (2022) employed an integrated analysis of carbon dioxide and oxygen concentrations which identified data offsets in float pH and derived $pCO_2$ data. Wu and Qi (2022) revealed an inconsistency between ship and BGC float-based $pCO_2$ in the Drake Passage region. Wimart-Rousseau et al. (2023) scrutinized the existing float pH correction process and found it insufficient, suggesting that a second reference point nearer to the surface would improve estimates.

Given the questions raised by these studies, a basin-scale quantitative examination of differences between float based $pCO_2$ and ship-based (Global Data Analysis Project: GLODAP) $pCO_2$ data proposed. Here we present the results of such a comparison in the SO. We apply various analytical tools such as $pCO_2$ vs. depth plots, $pCO_2$ vs. $O_2$ plots and plots which compare coupled deviations of $CO_2$ and $O_2$ concentrations from saturation (CORS plots as proposed by Wu et al. (2022)) to identify any discrepancies between the two datasets across 0-2000 m. Our analysis accounts for several factors that could

potentially affect these differences, ultimately concluding that issues with float pH data quality and its use for calculating $pCO_2$ contributed at least in part to the observed discrepancies.

## 2 Materials and Methods

### 2.1 Data sources

In this study, we focus on the SO delineated south of 50°S latitude, following Wu et al. (2022), and the surface layer is

defined as depths shallower than 100m, due to the deeper mixed layers in this region (Dong et al., 2008). We utilised comprehensive datasets comprising both float (over 300,000 sampling points) and ship (over 30,000 sampling points) measurements collected within the region (Fig. 1).



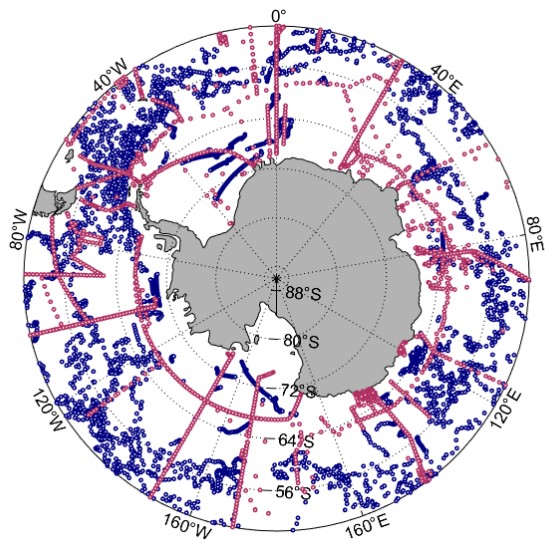

**Figure 1.** Location of ship data (red) and float data (blue) in the SO region.

### 2.1.1 BGC Argo float data

The BGC Argo data (NetCDF formatted version, accessed on 02 April 2023) were downloaded from https://data-argo.ifremer.fr/. We used "MATLAB toolbox for accessing and visualizing Argo data" (Frenzel, 2022) to select all 275 floats that entered the SO region in the period 2014 to 2023. The final float data (Matlab formatted version) used in this research includes: ID, longitude, latitude, depth, date, temperature, salinity, pH(in-situ), nitrate, oxygen and quality control

flags of pH(in-situ), nitrate, oxygen. All the BGC Argo data used in this study was delayed mode (calibrated by the Argo Global Data Assembly, Centers GDAC). The quality control (QC) flag value of "1" for the BGC Argo floats means that data quality is good according to the GDAC. Only the data where all of nitrate (to calculate TA), oxygen and pH had a QC flag of "1" were selected for our research (Schmechtig and Thierry, 2016).

### 2.1.2 GLODAP (ship) data

The GLODAP (ship) data were obtained from (https://www.glodap.info, last access: 05 October 2022). The GLODAPv2.2022 dataset contains almost 1.4 million internally-consistent samples of biogeochemical variables globally, collected on 1085 cruises during the period 1972–2021 (Lauvset et al., 2022). The final ship data used in this research includes: longitude, latitude, depth, date, temperature, salinity, dissolved inorganic carbon (DIC), TA, oxygen, silicate, phosphate and respective World Ocean Circulation Experiment (WOCE) flags. The WOCE flags were an indication of

whether the corresponding data is reliable. Only bottles where all of DIC, TA and oxygen were acceptable (WOCE flag 2) were selected for our research (Lauvset et al., 2021).



## 2.2 $pCO_2$ calculations

The MATLAB v3.1.2 of CO2SYS (Van Heuven et al., 2011, Sharp, 2023) was used to calculate $pCO_2$ using the carbonate system constants of Lueker et al. (2000), to ensure consistency with the previous studies (Williams et al., 2017, Bushinsky et al., 2019, Gray et al., 2018).For the ship data, $pCO_2$ was calculated from DIC and TA. For float data, $pCO_2$ was calculated from measured pH and algorithm-estimated TA. For the float data, the data processing was therefore divided into two steps: (1) estimation of TA and, (2) calculation of $pCO_2$ from estimated TA and observed pH (Williams et al., 2017). TA was estimated using the LIAR algorithm (Carter et al., 2018). We supplied the LIAR algorithm with the 3 necessary inputs: longitude, latitude and depth, and 4 additional parameters measured by the floats: temperature, salinity, nitrate and oxygen. The values of silicate and phosphate have a negligible effect on the calculation of $pCO_2$ from pH and TA (Williams et al., 2017) and were always entered as zero for float $pCO_2$ calculations. For ship $pCO_2$ calculations ($pCO_2$ from DIC and TA), on the other hand, silicate and phosphate have non-negligible effects and were taken from GLODAP data files (Lueker et al., 2000).

The float pH data recommended for researchers undergoes a calibration procedure at the data center: for each profile, the pH measured by the float at a depth of 1500, is compared to an expected value calculated from ship data. As properties are assumed to remain relatively constant over time, any pH difference detected is subtracted from the entire float profile (Maurer et al., 2021). The pH correction process for floats should result in a more accurate alignment between float and ship $pCO_2$ at depths below 1500 meters, as compared to data at other depths. The float pH data we used had been corrected in this way.

An additional adjustment is recommended when calculating $pCO_2$ from float pH (Williams et al., 2017) , that we applied is summarize thus: (1) float measured pH (in-situ) is converted to pH (25°C, 0 dbar). (2) Using equation (1) and the 1500m pH (25°C, 0 dbar), a correction is calculated for each profile. (3) This is added to each pH value in the profile before finally calculating $pCO_2$.

$$pH\ adjustment = -0.034529 \times pH(25°C) + 0.26709 \quad (1)$$

## 2.3 Approaches for interrogating differences in $pCO_2$

Three approaches were employed to investigate the disparity between float $pCO_2$ and ship $pCO_2$: $pCO_2$-depth plots, $pCO_2$-$O_2$ plots and CORS plots. The $pCO_2$-depth plot effectively illustrates systematic variations across the entire profile, making it an ideal method for elucidating differences in $pCO_2$ at different depths. The $pCO_2$-$O_2$ plots and CORS plots are based on the biogeochemical relationship between $CO_2$ and $O_2$. Previous research has demonstrated that oxygen sensor data from floats has the highest rate of 'good' data return (100%) while the pH sensors have a lower rate of good data return (88%) (Johnson et al., 2017). In crossover comparisons with GLODAPv2, the oxygen data measured by floats was found to exhibit a strong correlation with the corresponding GLODAPv2 ship-derived data, displaying a consistent 1:1 relationship (Johnson et al., 2017). However float pH data exhibited notable deviations from GLODAPv2 data, and the expected 1:1 relationship



(Johnson et al., 2017, Huang et al., 2023), with the addition of anthropogenic $CO_2$ in surface suggested as a potential cause.

Given that float-measured $O_2$ data exhibits a higher likelihood of accuracy compared to float-derived $pCO_2$, it is not affected by ocean acidification-related effects, and that they are both simultaneously involved in many processes (gas exchange, photosynthesis, respiration), $O_2$ data from floats can serve as a robust constraint for checking float $pCO_2$ values.

Taking into account the significant influence of surface $pCO_2$ on air-sea fluxes and the use of data from 1500 m in the calibration process, we calculated two metrics of particular interest: (1) the mean difference in $pCO_2$ values at the surface

(averaging over all data from 0 - 100 m, referred to as $\Delta pCO_{2, Surface}$) and (2) the mean difference in $pCO_2$ values in the deep ocean (averaging over all data from 1500-2000 m, referred to as $\Delta pCO_{2, DeepOcean}$).

### 2.3.1 $pCO_2$-depth

We use ship-float $pCO_2$ data comparisons between surface (depths <100m) and deep (1500-2000 m) waters to qualitatively assess the degree of overlap and deviation, with outputs binned into 100 m vertical intervals to examine depth-dependent

differences.

### 2.3.2 $pCO_2$-$O_2$

Similarly, we compare $pCO_2$ outputs against in situ oxygen levels, binning outputs into 10 μmol/kg oxygen intervals to examine potential disparities between them.

### 2.3.3 Carbon and oxygen relative to saturation (CORS)

The CORS method is an improved $CO_2$-$O_2$ analysis technique which involves comparing the deviations of $O_2$ and $CO_2$ concentrations to their respective saturation levels. Following Wu et al. (2022), we adopted an identical approach for both $O_2$ and $CO_2$ and proceeded to compare the dissolved concentrations of $O_2$ and $CO_2$ ([$O_2$] and [$CO_2$] respectively) in surface seawater with their respective saturation values. The saturation values represent the points at which the net air-sea gas exchange rate would be zero for each gas. Differences from saturation are calculated following:

$$\Delta O_2 = \left[ O_{2,observed} \right] - \left[ O_{2,saturation} \right] \ (2)$$

$$\Delta CO_2 = \left[ CO_{2,observed} \right] - \left[ CO_{2,saturation} \right] \ (3)$$

The saturation concentration of $O_2$ was calculated using Garcia & Gordon's equation (Garcia and Gordon, 1993, Garcia and Gordon, 1992); however, while they calculated the $O_2$ saturation concentration at an assumed 1 atm of atmospheric pressure, we instead used the local in-situ sea level pressure (SLP, usually < 1 atm in the SO) as per the below:

$$\left[ O_{2,saturation}^{1atm} \right] = K \times pO_2^{1atm} = K \times xO_{2,air} \times (P_{1atm} - P_{Sw}) \ (4)$$

$$\left[ O_{2,saturation}^{SLP} \right] = K \times pO_2^{SLP} = K \times xO_{2,air} \times (P_{SLP} - P_{Sw}) \ (5)$$

Here, the '1atm' and 'SLP' indicate two different pressures. K is the solubility of oxygen, $O_{2,saturation}^{1atm}$ is the result based on the Garcia & Gordon's equation (Garcia and Gordon, 1993, Garcia and Gordon, 1992). $O_{2,saturation}^{SLP}$ is the result based on



our modified equation. $P_{Sw}$ is the water vapor pressure calculated from the surface temperature and salinity (Weiss and Price, 1980b). $P_{SLP}$ is taken from the monthly gridded climate data downloaded from National Oceanic and Atmospheric Administration (NOAA) (Gridded Climate: NOAA Physical Sciences Laboratory, last access: 10 July 2022). Combining equations (4) and (5):

$$[O_{2,saturation}^{SLP}] = [O_{2,saturation}^{1atm}] \times (P_{SLP} - P_{Sw})/(P_{1atm} - P_{Sw}) \quad (6)$$

$[CO_{2,saturation}]$was calculated with Henry's equation. $([CO_{2,observed}] = K_H \times pCO_{2,equilibrium})$, where $pCO_{2,equilibrium}$ is the partial pressure of $CO_2$ in seawater at equilibrium with atmospheric $CO_2$; it was calculated according to the equation:

$$pCO_{2,equilibrium} = xCO_{2,air} \times (P_{SLP} - P_{Sw}) \quad (7)$$

where the $xCO_{2,air}$ is the mole fraction of carbon dioxide (ppm) in dry atmosphere. The monthly mean atmospheric $xCO_{2,air}$ values from the monitoring site at Palmer Station, Antarctica were used for the SO (downloaded from https://www.esrl.noaa.gov/gmd/ccgg/trends/, last access: 14 July 2022). The solubility $(K_H)$ of carbon dioxide is calculated from the formula proposed by Weiss (1974).

## 3 Results

### 3.1 $p$CO$_2$ differences at different depths

Initially, scatter plots were generated to compare the $p$CO$_2$ data obtained from floats and ships at different depths. Fig.2 shows a difference between the $p$CO$_2$ values derived from the floats and those acquired from the ships. Generally, the ship data and float data exhibit a substantial overlap, although a fraction of the float data has higher $p$CO$_2$ values than the ship data from the same depths.

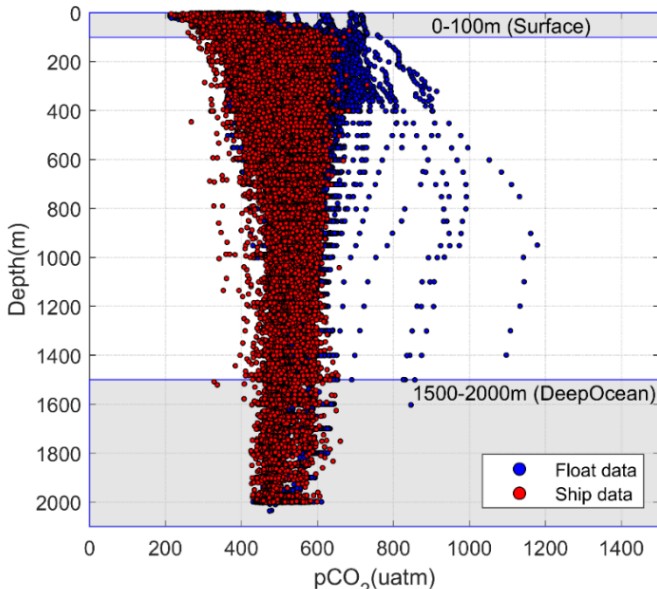





**Figure 2:** Scatterplot of $pCO_2$ at different depths in the Southern Ocean. Float data (blue) is plotted first and ship data (red) second (Supplement Fig. S1 shows the same plot but with ship data plotted first).

The mean difference plot (Fig. 3) reveals two notable observations. Firstly, the $pCO_2$ values derived from the float data are, on average, higher than the ship data at most depths. Secondly, the discrepancy between the float and ship data is particularly pronounced at depths shallower than 1500 m.

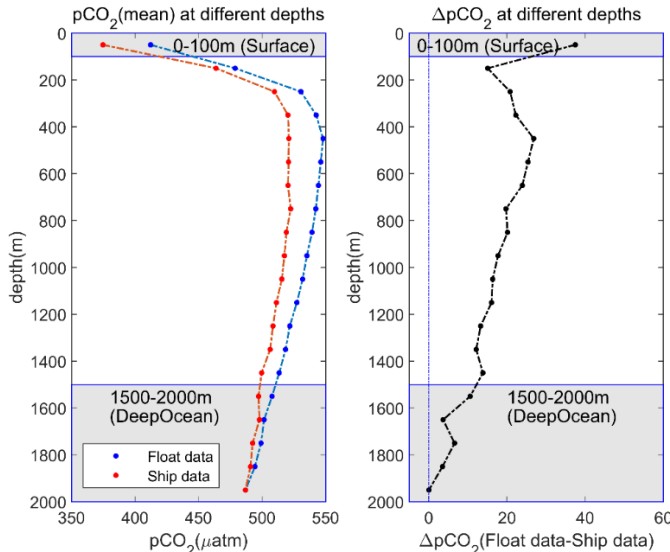

**Figure 3:** $pCO_2$(mean) at different depths. The left panel shows the average float and ship $pCO_2$ at different depths; the right
panel shows the difference between the two (float $pCO_2$ minus ship $pCO_2$).

The values of the two metrics of particular interest (section 2.3) are $\Delta pCO_{2,Surface}$ = 37.5 µatm and $\Delta pCO_{2,DeepOcean}$ = 4.9 µatm. These act as benchmarks for later analysis and discussion. As is apparent in Fig. 2, some float data points exhibit unusually high $pCO_2$ values (over 750 µatm), despite being corrected and marked as good by the GDAC. We checked for the effect of removing these possibly anomalous data; however, because we were not certain that the data are incorrect, and because their
removal had minimal effect on the $pCO_2$ averages (no more than 0.1 µatm), we did not exclude them from our analyses and plots.

### 3.2 $pCO_2$ differences from $pCO_2$-oxygen relationships

O$_2$ increases/decreases are often accompanied by $pCO_2$ decreases/increases because of photosynthesis and respiration, leading to a negative correlation between the two (Fig. 4). Here, it is evident that while the O$_2$ values exhibit similarity, the
$pCO_2$ derived from float data demonstrates an overall higher magnitude in parameter space than that obtained from ship data.



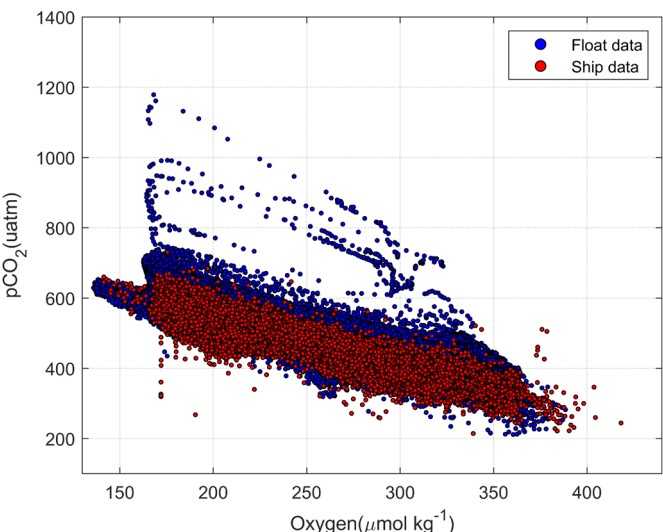

**Figure 4:** Scatterplot of $p\mathrm{CO_2}$ against oxygen (ship data and float data) in the SO region. (Float data (blue) is plotted first and ship data (red) second (Supplement Fig. S2 shows the same plot but with ship data plotted first, to avoid the effects of masking)

We further calculated the average $p\mathrm{CO_2}$ values within different oxygen intervals (ranging from 150 to 360 μmol/kg) and examined the differences between them (Fig. 5). The average $p\mathrm{CO_2}$ difference was calculated to be 19.3 μatm (obtained by subtracting ship data from float data). Our analyses indicate a consistent and significant systematic disparity between the $p\mathrm{CO_2}$ values obtained from float data and ship data, with consistently higher float $p\mathrm{CO_2}$ values within each oxygen interval.

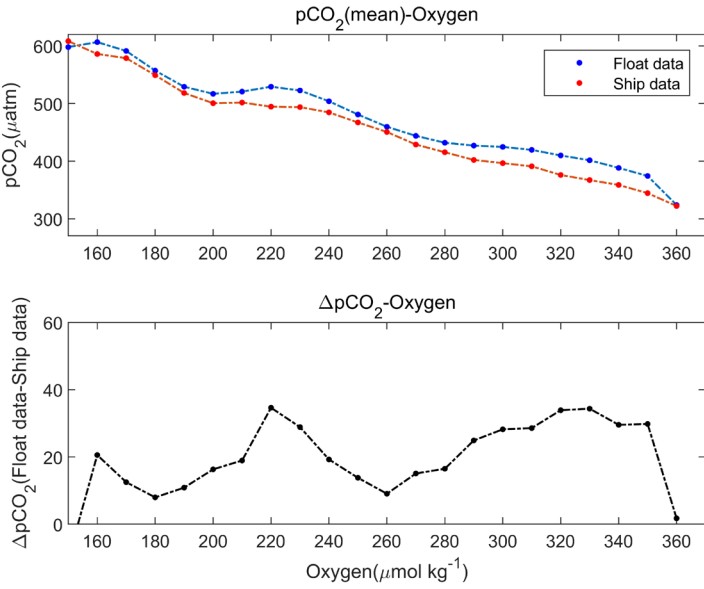





**Figure 5:** $p$CO$_2$ (mean) against oxygen of float and ship data (above); the difference ($\Delta p$CO$_2$, below) is calculated as float $p$CO$_2$ minus ship $p$CO$_2$.

### 3.3 CORS plots of ship data and float data

Fig. 6 shows a CORS plot of both ship data and float data from the Southern Ocean. The pattern of float data is comparable to that of ship data: a large number of data points are situated in the top-left quadrant, i.e., supersaturation of CO$_2$ along with
undersaturation of O$_2$. This pattern can be attributed to upwelling of CO$_2$-rich and oxygen-poor deep waters, where the deep-water composition has been changed by remineralization of sinking organic matter.

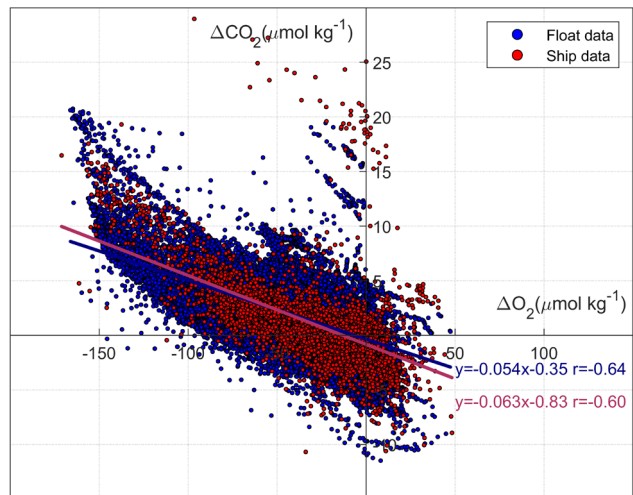

**Figure 6:** CORS plot from surface float data (blue) and ship data (red). The blue line is the best-fit line to float data and the pink line is the best-fit line to ship data. r is the associated Pearson correlation coefficient. Float data (blue) is plotted first
and ship data (red) second, masking the locations of many float data.

In situations where biogeochemical and physical processes like photosynthesis and upwelling strongly influence the system, the deviations of O$_2$ and CO$_2$ from atmospheric equilibrium demonstrate a coupled behavior, as photosynthesis and remineralization affect carbon and oxygen in contrasting directions, according to a stoichiometric ratio. We find different CORS plot y-intercepts for float data and for ship data (Fig 6): the ship data based y-intercept is –0.83 μmol kg$^{-1}$, while the
float data based y-intercept is -0.35 μmol kg$^{-1}$, a difference of 0.48 μmol kg$^{-1}$ (float data minus ship data). Converting the offset in [CO$_2$] (μmol kg$^{-1}$) to $p$CO$_2$ (μatm), assuming average values for sea surface temperature (1°C) and salinity (35), gives a mean difference in surface $p$CO$_2$ of 7.7 μatm.



## 4 Discussion

The results in Fig. 3, Fig. 5 and Fig. 6 show a notable disparity between $pCO_2$ values derived from float data and those
obtained from ship data. We now examine potential causes of the differences other than data error. There are several factors
that could potentially influence the differences in $pCO_2$ values: (i) spatial sampling bias between the two databases, (ii)
seasonal bias in the ship-collected data, (iii) errors in estimated alkalinity, (iv) errors in carbonate system constants, and (v)
higher levels of anthropogenic $CO_2$ in float data. Only by excluding these factors can we conclude that the observed
deviation in $pCO_2$ is likely to be due to problems with the quality of the $pCO_2$ data from the floats.

### 4.1 Impact of different factors on disparities

### 4.1.1 Spatial sampling bias between two databases

To exclude the potential impact of sampling area variations, we compare float-measured $pCO_2$ and ship-measured $pCO_2$ data
within specific regions. In particular, we focus on the Drake Passage region (55°S to 65°S, 55°W to 70°W) as a
representative case study (Fig. 7) (Wu and Qi, 2022, Wu et al., 2022). Comparing data only from within this specific region
(where there are large amounts of both ship and float data) excludes any confounding effects arising from variations in
sampling locations.

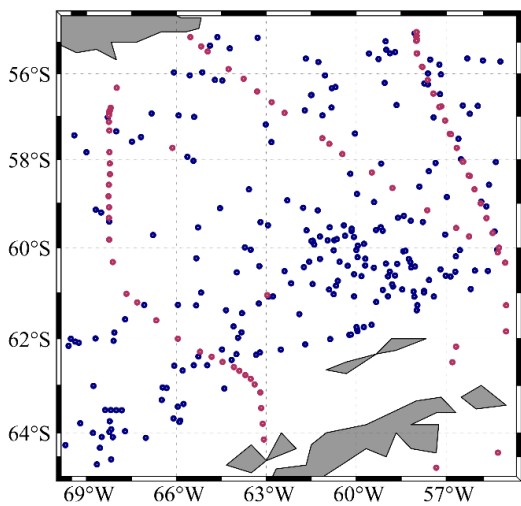

**Figure 7:** Location of ship data (red) and float data (blue) in the Drake Passage region.

In line with our analysis of the complete dataset encompassing the Southern Ocean, we computed the average $pCO_2$ values at
various depths. Within the Drake Passage region (1166 ship data points and 16137 float data points) we observed a
$\Delta pCO_{2,Surface}$ value of 37.2 µatm, accompanied by a $\Delta pCO_{2,DeepOcean}$ value of -16.8 µatm (Fig. 8), compared to 37.5 and 4.9
µatm for the SO as a whole (section 3.1). A similar analysis in the region around the prime meridian, near Antarctica (70°S-
65°S, 10°W-10°E) (1226 ship data points and 6818 float data points), produced $\Delta pCO_{2,Surface}$ = 45.4 µatm and $\Delta pCO_{2,DeepOcean}$





= 1.7 µatm (Supplement Fig. S3 and S4). The persistence of substantial differences in both regions indicates that spatial variability alone does not entirely account for the disparities in $pCO_2$ between the two datasets.

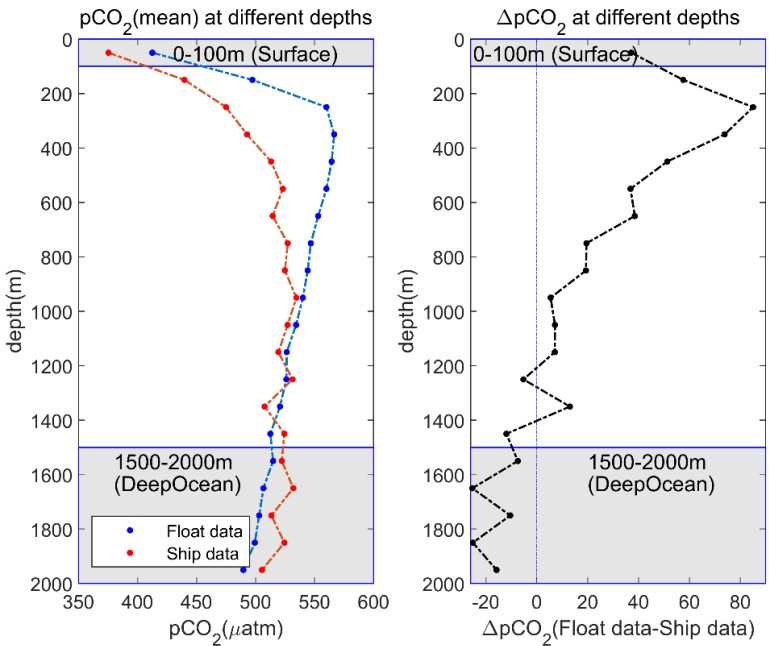

**Figure 8:** $pCO_2$ (mean) at different depths (left), and difference between float $pCO_2$ minus ship $pCO_2$ at different depths (right) in Drake Passage region.

### 4.1.2 Seasonal bias in ship data

To mitigate the potential influence of sparse winter sampling conducted by ships, we carefully assessed the distribution of ship data across different months (Supplement Fig. S5). Float data is evenly distributed over the seasons, while ship data is mainly concentrated in summer. Consequently, we specifically selected ship and float data from November to April, omitting winter data (May to October) from both datasets. This eliminated the possible confounding effect of seasonal variations in sampling intensity. The resulting average differences in $pCO_2$ values are $\Delta pCO_{2,\text{Surface}}$ = 32.4 µatm and

$\Delta pCO_{2,\text{DeepOcean}}$ = 6.6 µatm. $\Delta pCO_{2,\text{Surface is}}$ reduced by 5.1 µatm, while $\Delta pCO_{2,\text{DeepOcean}}$ is increased by 1.7 µatm. By removing seasonal bias, we were able to slightly reduce the sea surface $pCO_2$ difference, but the finding of an overall higher $pCO_2$ in the floats did not change.

### 4.1.3 Errors in estimated alkalinity

In order to examine the difference between TA estimated from float data and TA measured from ships, the average

difference at different depths between float TA and ship TA was calculated in the same way as for $pCO_2$ data. TA from ships was on average 3.6 µmol kg$^{-1}$ higher than TA estimated from floats. This offset was used to correct the float TA and $pCO_2$




was calculated again using the corrected float TA and the measured pH, yielding a new $\Delta p\mathrm{CO}_{2,\mathrm{Surface}}$ of 38.1 µatm and $\Delta p\mathrm{CO}_{2,\mathrm{DeepOcean}}$ of 5.6 µatm. Therefore, the $p\mathrm{CO}_2$ differences between the two databases are not due to TA errors, confirming previous studies that reached similar conclusions (Williams et al., 2017, Gray et al., 2018, Bushinsky et al., 2019, Wu et al., 2022).

### 4.1.4 Errors in carbonate system constants

In the calculation of $p\mathrm{CO}_2$ for floats and ship, we used the Lueker et al. (2000) carbonate dissociation constants in the CO2SYS program. Some recent work has shown that current estimates of the carbonate system constants appear inaccurate in cold ocean regions (Woosley and Moon, 2023, Sulpis et al., 2020). We recalculated the $p\mathrm{CO}_2$ values with two other sets of constants, Sulpis et al. (2020) and Roy et al. (1993). With Sulpis et al.'s constant, the updated $\Delta p\mathrm{CO}_{2,\mathrm{Surface}}$ value of 39.7 µatm and $\Delta p\mathrm{CO}_{2,\mathrm{DeepOcean}}$ value of 7.3 µatm reflect a slightly increase, while, conversely, employing Roy et al.'s constants yielded a new $\Delta p\mathrm{CO}_{2,\mathrm{Surface}}$ value of 31.5 µatm and $\Delta p\mathrm{CO}_{2,\mathrm{DeepOcean}}$ value of -2.0 µatm, a reduction in both variables. It is evident from the findings that employing different carbonate system constants in the calculations can alter the differences. However, neither set removes the significant discrepancy between ship and float $p\mathrm{CO}_2$ values. In order to ensure consistency and enable meaningful comparisons with previous studies, we have used Lueker et al.'s (2000) constants as the chosen carbonate system constants throughout the rest of this study.

### 4.1.5 Higher levels of anthropogenic $CO_2$ in float data

The $p\mathrm{CO}_2$ in the ocean responds to the rising trends in atmospheric carbon dioxide concentrations due to the burning of fossil fuels (Bates et al., 2014). Float data has, on average, been collected more recently than ship data, and so the water the floats sampled can be expected to contain higher levels of anthropogenic $CO_2$ (Supplement Fig. S6). To prevent temporal $p\mathrm{CO}_2$ trends being mistaken for discrepancies, we normalized both float and ship surface $p\mathrm{CO}_2$ to the reference year of 2005, where "surface data" is defined as data collected in the uppermost 100m (de Boyer Montégut et al., 2004). The normalization is based on assuming that surface $p\mathrm{CO}_2$ tracks atmospheric $p\mathrm{CO}_2$ (Feely et al., 2008, Wu et al., 2019). First, we calculated the variation in the molar fraction of $CO_2$ in the atmosphere ($x\mathrm{CO}_{2,\mathrm{air}}$) from the reference year 2005 to the year of data collection.

$$\Delta x CO_{2,air} = x CO_{2,air}^{Year} - x CO_{2,air}^{2005} \quad (8)$$

where the superscript "Year" refers to the year of data collected and the superscript "2005" refers to the reference year. The globally annual average $x\mathrm{CO}_{2,\mathrm{air}}$ data was downloaded from (https://www.esrl.noaa.gov/gmd/ccgg/trends/, last access: 18 April 2023). We used mean values of float in-situ surface ocean temperature (T) and salinity (S) data to calculate humidity (pH$_2$O) data (Weiss and Price, 1980a),

$$\ln pH_2O = 24.4543 - 67.4509 \left(\frac{100}{T}\right) - 4.8489 \ln\left(\frac{T}{100}\right) - 0.000544 S \quad (9)$$

then converted $\Delta x\mathrm{CO}_{2,\mathrm{air}}$ into $\Delta p\mathrm{CO}_{2,\mathrm{air}}$ (Takahashi et al., 2014, Weiss and Price, 1980a).

$$\Delta pCO_{2,air} = \Delta x CO_{2,air} \times (1 - pH_2O) \quad (10)$$





Based on the assumption that $\Delta pCO_{2,\text{surface}}$ is equal to $\Delta pCO_{2,\text{air}}$, the surface $pCO_{2,\text{surface}}$ normalized to reference year 2005 was calculated as

$$pCO_{2,surface}^{2005} = pCO_{2,surface}^{Year} - \Delta pCO_{2,surface} = pCO_{2,surface}^{Year} - \Delta xCO_{2,air} \times (1 - pH_2O) \ (11)$$

where $pCO_{2,surface}^{Year}$ is the in-situ $pCO_2$ at the time of data collection, whether by float or ship. After normalization of the sea surface $pCO_2$ data, the new $\Delta pCO_{2,\text{Surface}}$ is 13.1µatm and $\Delta pCO_{2,\text{DeepOcean}}$ is still 4.9µatm. The temporal $pCO_2$ trends and the ongoing accumulation of anthropogenic $CO_2$ in surface waters can thus explain a sizeable portion of the difference but does not eliminate it completely.

**4.2 Adjusted $pCO_2$ differences after taking multiple factors into account**

Each of these factors individually influenced the variance results to some degree, but none can independently fully explain the observed differences in $pCO_2$ (Table 1). However, it is worth considering whether the inclusion of all of these influences together would yield a different result. Since the spatial bias (4.1.1) was obtained by separately calculating the data for the different regions, we do not include it here in the adjustment to the Southern Ocean data, but rather as a separate control

analysis. For carbonate system constants (4.1.4), the discrepancies either increase or decrease depending on which alternative set of constants is used, and we decided to stay with the constants of Lueker et al. (2000) in order to maintain consistency with other studies. Therefore, no correction was made for this. Here we take the seasonal bias (4.1.2), errors in estimated TA (4.1.3), and anthropogenic $CO_2$ influence (4.1.5) into account and recalculate the $pCO_2$ differences.

**Table 1**

*Possible reasons for the $pCO_2$ discrepancies and their magnitudes after calculations to exclude their influence (changes in value from the initial analysis are shown in brackets).*

| Possible reasons of observed $pCO_2$ discrepancy | $\Delta pCO_{2,\text{DeepOcean}}$ (after adjustment) | $\Delta pCO_{2,\text{Surface}}$ (after adjustment) |
|---|---|---|
| Initial analysis (no adjustments) | **4.9** | **37.5** |
| 1. Spatial sampling bias between two databases (top line Drake Passage, bottom line region around the prime meridian) | -16.8 (-21.7) µatm/ 1.7 (-3.2) µatm | 37.2 (-0.3) µatm/ 45.4 (+7.9) µatm |
| 2. Seasonal bias in ship-collected data | 6.6 (+1.7) µatm | 32.4 (-5.1) µatm |
| 3. Errors in estimated alkalinity | 5.6 (+0.7) µatm | 38.1 (+0.6) µatm |
| 4. Errors in carbonate system constants (top line Sulpis et al., 2002; bottom line Roy et al., 1993) | 7.3 (+2.4) µatm/ -2.0 (-6.9) µatm | 39.7 (+2.2) µatm/ 31.5 (-6.0) µatm |
| 5. Higher levels of anthropogenic $CO_2$ in float | - | 13.1 (-24.4) µatm |



| data (surface only) | | |
|---|---|---|
| 2, 3 & 5 combined | **7.4** (+2.5) µatm | **9.4** (-28.1) µatm |


Excluding seasonal, alkalinity and anthropogenic $CO_2$ effects results in updated estimates of $\Delta pCO_{2,\text{DeepOcean}}$: 7.4 µatm, and $\Delta pCO_{2,\text{Surface}}$: 9.4 µatm (Fig. 9). The adjusted average difference between the various oxygen intervals was calculated to be 10.0 µatm (Supplement Fig. S7). In the case of CORS plots, considering that $\Delta CO_2$ is the difference between the observed concentrations relative to the saturated concentration at the time of sampling, normalizing the $pCO_2$ to year 2005 isn't

required. Here we take the factors 2 & 3 (from Table 1) into account when constructing an adjusted CORS plot. The final y-axis intercept difference is 0.40 µmol kg$^{-1}$. As noted above, the offset in y-intercept ([$CO_2$], in µmol kg$^{-1}$) can be converted to a deviation in $pCO_2$ (µatm) by assuming a temperature of 1℃ and salinity of 35; doing this yields an adjusted surface $pCO_2$ difference of 6.4 µatm (Supplement Fig. S8).

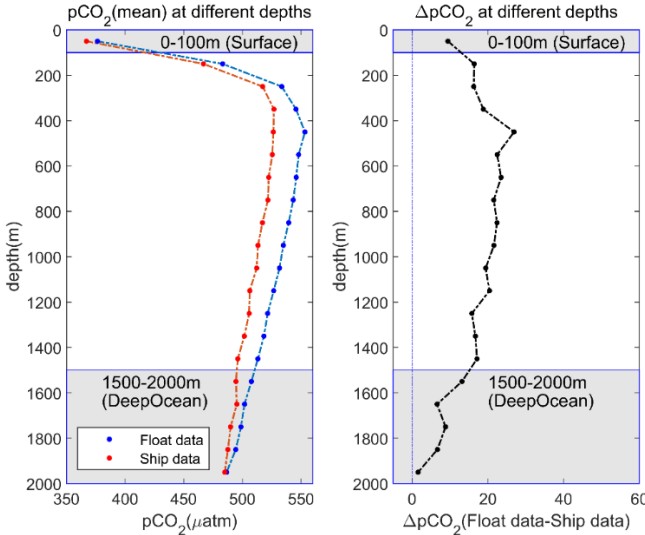

**Figure 9:** Adjusted $pCO_2$ (mean) (incorporating factors 2, 3 & 5 in Table 1) at different depths (left), and difference between float $pCO_2$ minus ship $pCO_2$ at different depths (right) in the SO.

**4.3 Evaluation by comparison to SOCAT data**

An alternative source of ship-collected $pCO_2$ data, for surface waters only, is the Surface Ocean $CO_2$ Atlas (SOCAT) data (Bakker et al., 2016). $pCO_2$ values in SOCAT are from direct measurements, believed to be highly accurate (<2 µatm) and

reliable. We compared float $pCO_2$ to SOCAT $pCO_2$ in the region shown in Fig. 10 (Munro et al., 2015a). After accounting for factors 2 & 5 (Table 1), we obtained a $pCO_2$ discrepancy (float minus SOCAT) of 8.3 µatm. The result is close to the





discrepancy obtained from previous comparison of the float data and GLODAP data. This is strong evidence that there is inaccuracy in the float data $pCO_2$.

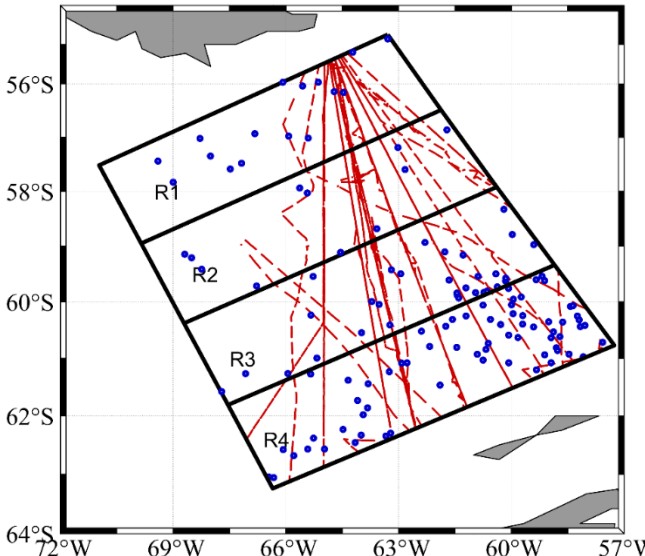

**Figure 10:** Location of SOCAT data (red) and float data (blue) in the Drake Passage region. The divisions R1-R4 are those used by Munro et al. (2015).

Our rigorous analysis and assessment, as summarized in Table 1: we account for several of the above factors simultaneously, and obtain a final $pCO_2$ difference. To determine the effects of spatial sampling bias, we apply the same analysis to specific regions as to all the SO and the discrepancy remain clear in regional analyses. To correct for seasonal bias, we subtract 5.1

µatm from surface values and add 1.7 µatm to deep values. To compensate the errors in estimated alkalinity, we add 3.6 µmol kg$^{-1}$ to the float TA data in order to obtain the final TA value. For the anthropogenic $CO_2$ effect, we normalize both the ship $pCO_2$ and float $pCO_2$ to the reference year of 2005 to remove the temporal $pCO_2$ trends, assuming (this can only be partially correct, given the long timescales of equilibration of $CO_2$ in the SO (Jones et al., 2014)) that air-sea gas exchange has run to completion in all surface waters. For carbonate system constants, the discrepancies either increase or decrease

depending on which alternative set of constants is used, and we decided to stay with the constants of Lueker et al. (2000) in order to maintain consistency with other studies.

### 4.4 Final $pCO_2$ differences and potential sources

Discrepancies remain even after the several possible explanatory factors (2, 3, & 5 in Table 1) were applied simultaneously to the whole of the SO. The final $\Delta pCO_{2,\text{Surface}}$ value stands at 9.4 µatm, while the final $\Delta pCO_{2,\text{DeepOcean}}$ value is 7.4 µatm.

(For the factor 1, we applied the adjustedment of factor 2, 3, 5 to data in Drake Passage region and region around the prime meridian, obtained the $\Delta pCO_{2,\text{Surface}}$ values are 11.4 µatm and 9.3 µatm respectively. They served as a control and are not included in the final consideration of the overall deviation of the Southern Ocean.) Lastly, we employed hypothesis testing



to enhance our findings. Firstly, we examined the distribution of both ship and float $pCO_2$ adjusted data (Supplement Fig. S9), utilizing Kolmogorov-Smirnov Testing to assess the normality of the two databases individually. The results rejected (p

< 0.01) the null hypothesis that either ship or float $pCO_2$ follows a normal distribution. Consequently, we employed Mann-Whitney U Testing to evaluate the consistency between adjusted ship $pCO_2$ and float $pCO_2$ data (Supplement Fig. S10). These tests rejected (p < 0.01) the null hypothesis of no difference between ship and float data. Based on the outcomes of these hypothesis tests, it can be concluded that, even after accounting for all factors corrected, there still exists a significant offset between ship $pCO_2$ and float $pCO_2$. It should be noted that although we find discrepancies between average values, we

do not assert that all float-based $pCO_2$ data are erroneous; rather, our regional and overall assessments underscore the existence of uncertainties within the float data.

It is necessary to be aware that this final $\Delta pCO_{2,\text{Surface}}$ value may be an underestimate. The largest adjustment we applied (reducing $\Delta pCO_{2,\text{Surface}}$ by 24.4 µatm) was the normalization of $pCO_2$ to a reference year of 2005, to eliminate the impact of increasing anthropogenic carbon over time. This may, however, be an overcorrection, because it assumes that air-sea gas

exchange has run to completion in all surface waters. The equilibration of $CO_2$ is estimated to take many months in the SO (Jones et al., 2014); for this reason, it is likely that some surface waters were not at equilibrium at the time of measurement. The overall saturation level is difficult to quantify, but a modelling study suggests considerable deviations from saturation (Fig. 8(c) of Jones et al. (2014)). Widespread ocean-atmosphere imbalances would undoubtedly increase the value of the final discrepancy. Therefore, we consider our $\Delta pCO_{2,\text{Surface}}$ calculation of 9.4 µatm to be a conservative estimate.

**4.5 Implications**

Returning to our initial question, correctly defining $pCO_2$ at the sea surface is crucial in constraining the global oceanic carbon sink, making the $\Delta pCO_{2,\text{Surface}}$ parameter highly meaningful for $CO_2$ flux estimates. This result of a definite, non-negligible discrepancy has the following implications: (1) float-based $pCO_2$ estimates are biased high compared to other observations in the SO; (2) this bias can potentially explain the mismatch between float-based $CO_2$ air-sea flux estimates and

those from other sources (e.g. Long et al., 2021); (3) our analyses suggests that the overestimation of surface $pCO_2$ by floats could be roughly within the uncertainty estimate of Williams et al. (2017) of ±2.7% (i.e., ±11 µatm), although it could also be higher.

Such a high bias implies that the current calculation scheme for float pH and by extension $pCO_2$ has quality problems in agreement with (Wimart-Rousseau et al., 2023) and others. In order to maximize the utility of carbon measurements on

floats, a comprehensive reconsideration of all factors contributing to uncertainties in float $pCO_2$ is recommended. Williams et al. (2017) grouped sources of uncertainty in float $pCO_2$ values into three main categories: (1) pH sensors, (2) alkalinity estimates, and (3) carbonate system equilibrium constants. Future work should identify which out of these three categories is responsible for the majority of the $pCO_2$ biases and focus on reducing these uncertainties.

Gray et al. (2018) and Bushinsky et al., (2019) previously implemented adjustment to float $pCO_2$ corresponding to a 4 µatm

reduction across the Southern Ocean surface layer, considering the differences with nearby ship's data. Other studies (e.g.





Huang et al., 2023) did not implement an adjustment, following a recommendation by the U.S. Ocean Carbon and Biogeochemistry Program working group entitled the Ocean Carbonate System Intercomparison Forum (SI Appendix, Text S1.8) not to do so. Our results suggest that an adjustment should be applied, and that it should be larger than 4 µatm. The use of a 4 µatm reduction brought Gray et al.'s (2018) results closer to, albeit still significantly higher than, the flux calculated

from ship data and other database results (Long et al., 2021, Gray et al., 2018, Bushinsky et al., 2019). Considering the differences we have identified, the comprehensive adjustment of Southern Ocean float surface $pCO_2$ for the purpose of calculating air-sea flux is anticipated to yield a significantly greater $CO_2$ uptake estimate. Consequently, such an adjustment will most likely enhance the consistency of float calculations with other databases and promote a greater alignment of $CO_2$ flux estimates.

**5 Conclusions**

In this study, a significant inconsistency was found between float-based and ship-based $pCO_2$ in the Southern Ocean. Float $pCO_2$ values were found to be, on average, higher than ship $pCO_2$ values obtained at the same depths across the entire float profile. Supporting this conclusion, float $pCO_2$ also exhibits higher values overall when compared with oxygen relationships. Several alternative possible explanations were considered for the observed discrepancy in $pCO_2$: spatial sampling bias,

seasonal sampling bias, errors in TA, choice of carbonate system constants and higher levels of anthropogenic $CO_2$ in float data. These factors were found to change the value of the difference to some extent, but could not fully explain the observed discrepancy. Applying several corrections together left a discrepancy of 10 µatm in the top 100m, 7 µatm at depth. The conclusion is further supported by CORS analysis. Thus, we conclude that the $pCO_2$ discrepancies are due to float $pCO_2$ data quality issues. By incorporating this discrepancy to further refine the Southern Ocean surface $pCO_2$ data, a notable increase

in the estimated values of net air-sea $CO_2$ uptake is anticipated. This adjustment is expected to align the calculated flux results more closely with other independent datasets in Southern Ocean. Our results suggest that the calculation scheme of $pCO_2$ from float pH needs further refinement; all sources of uncertainty should be considered, in order to identify those processes that lead to the disparities obtained in this study. More accurate $pCO_2$ data derived from floats will be of great value in future analyses of Southern Ocean carbon uptake and release, and globally.


**Acknowledgments**

We acknowledge the efforts of the data providers and scientists who conducted data collection, quality control, and synthesis, which ultimately resulted in the creation of the datasets used in this study. We also appreciate the other researchers who

provided comments and constructive feedback on this work.



**Code/Data Availability**

All BGC-Argo data Data were collected and made freely available by the Global Ocean Biogeochemistry Array (GO-BGC)
Project funded by the National Science Foundation, Division of Ocean Sciences (NSF OCE-1946578), and by the
International Argo Program and the national programs that contribute to it. ([http://www.argo.ucsd.edu](http://www.argo.ucsd.edu), [http://ocean-ops.org](http://ocean-ops.org)).
The Argo Program is part of the Global Ocean Observing System. The GLODAPv2.2022 dataset was downloaded from
[https://glodap.info/](https://glodap.info/).     The monthly mean atmospheric $xCO_2$ values for observing site were obtained from
[ftp://aftp.cmdl.noaa.gov/data/trace_gases/co2/flask/](ftp://aftp.cmdl.noaa.gov/data/trace_gases/co2/flask/).

**Declaration of Competing Interest**

The authors declare that they have no known competing financial interests or personal relationships that could have
appeared to influence the work reported in this paper.

**Author contribution**

C.Z. and T.T. conceived the original idea and came up with methods of research. C.Z. performed the analytic calculations,
coding and writing. T.T. supervised the project. Y.W, P.B., D.S. and A.S provided significant comments and help specific in
coding, calculating and Data collection.

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

         *Science Bulletin***,** S2095-9273 (23) 00615.