# Peer review of "Comparing float $pCO_2$ profiles in the Southern Ocean to ship data reveals discrepancies"

_EGUsphere, 2023_

## Referee Comment (RC1)

**Comparing float *p*CO2 profiles in the Southern Ocean to ship data reveals discrepancies**

Summary: The authors compare the means of pCO2 data measured by ships (1972-2021) with pCO2 data derived from Argo floats (2014-2023) in the Southern Ocean. They find an increase in the float pCO2 values compared to the ship pCO2 and explain the mean difference by seasonality, trends in atmospheric CO2, differences in sampling location, errors in TA, and the choice of carbonate system constants. Consequently, they adjust the pCO2 values by removing the influence of these factors on the mean (e.g., normalizing the data to a reference year). They attribute the remaining difference in mean pCO2 to quality issues. While I appreciate the concept of comparing float pCO2 data with ship pCO2 data, I have major issues with this study. My greatest concerns are as follows:

1. **Content**: The study falls short in making a meaningful contribution to the existing knowledge base. It lacks the presentation of any novel findings. (Higher pCO2 observed in Argo float data than ship data → partially caused by seasonality, different sampling location etc). The conclusion of "bad data quality" appears inadequate given the methodology and is insufficiently discussed)
2. **Methodology**: I fail to understand the rationale behind comparing data from various time periods, seasons, and sampling locations in the first place, particularly when focusing solely on the mean values. In my opinion, this approach is simply not acceptable, as e.g. ocean biogeochemistry undergoes changes over time, leading to higher CO2 levels in more recent float data. While the authors acknowledge this in their later analyses presented in the discussion section, the results in the results section of the study are therefore not comparable. Additionally, the study does not quantify the sources of uncertainty in float pCO2 data, rendering the conclusions regarding data quality issues questionable.
3. **Structure**: Result sections 3.1, 3.2 and 3.3 should be merged as the subsections merely contain different plots. The discussion section comprises the presentation of additional analyses, thereby resembling more of a result section.
4. **Choice of visualization**: The content in Figure 2, 3, S1 as well as 4, 5, S2 could be merged (remove scatterplots, add error bars to line plots).
5. Authors doubt/question data quality without further arguments (l.200-206). After adjusting the means, they did not go into "float pH data quality issues". I would have appreciated a discussion on why the quality is perceived as poor and how it could be improved etc

**Minor, detailed comments**

l. 9: Please specify why/what role

l.12: It reads like "pCO2 estimates obtained… from ship-collected data" which would imply that pCO2 is also estimated from ship data (and not directly measured). Please rephrase it.

l. 14: Regarding the term "statistically significant differences," if not statistically significant, perhaps "substantial differences" might be more appropriate.

l.14: It might be beneficial to consider removing the phrase "A preliminary analysis, ignoring other factors," as it could prompt questions about the factors being ignored and why.

l.17 What "other factors"

Figure 1: Could you please specify the time period covered? Is it from 2014 to 2023?

l. 82: missing word ("is proposed"). Missing comma: "Here,"

l.87: The Method section appears incomplete and should be expanded to include the methods used to obtain the results presented in the discussion section.

l.101: I would suggest merging the sentence to "Only data where the parameters nitrate, oxygen, and pH had a quality control (QC) flag of "1" equivalent to good quality according to GDAC were used here (Schmechtig and Thierry, 2016)"

l.104: Were data from 1972-2021 used?

l. 115: missing blank space before "For the ship data"

l. 124: interesting word choice "recommended for researchers". Are there more data recommended for other groups?

l.124 What data center? Consider rephrasing or clarify whether it is necessary to include this detail. Maybe rephrase to "The float pH data are calibrated:" Or is the data center relevant?

l.128: Could you provide more context for choosing 1500m?

l. 130: Please remove blank space

l. 130-134: The sentence appears incomplete, and the bullet points merely reiterate the equation. It may be clearer to remove the bullet points and rephrase as follows: "We adjusted the in-situ pH measured by floats according to the method outlined by Williams et al., 2017, using the equation [equation]."

l.135: While "Interrogating" may be technically correct, it sounds very awkward/formal. You might want to opt for a more common verb like Instead investigating

l. 138 Perhaps consider using "visualizing" instead of "elucidating"

l. 143: Add a comma after "However",

l.144: missing word (is suggested)

l.188: Please consider removing the first sentence as it lacks meaningful information.

l.188: add blank space (Fig. 2)

Figure 2: The reader can't see the blue float data points underneath the red ship data (another plots with reversed order seems redundant).
One option I would suggest would be removing the black outline of the markers for better visibility and increasing the transparency. This would also make Supplement Figure S1 redundant. Since Figure 2 and 3 are closely related, I would suggest merging it. Add the lines from Figure 3 in Figure 2.
Option 2 (preferred option): plot mean/both lines (red-blue and black) in one plot and add error bars.

l.195: I would suggest merging the 3 sentences: We observe that mean pCO2 values derived from floats exceed those than from ships with highest differences occurring at depths shallower than 1500 m.

l.201-202: If the two values are that important, I would suggest adding them to the plot. I don't think it's necessary to announce how important they are here without explaining further. I would suggest removing both sentences.

l.202: I would suggest removing "as is apparent in Fig. 2" and just add "(Fig. 2)" at the end.

l.204: Why should we assume the data is incorrect simply because it exhibits high values? Shouldn't the flag 1 remove any data considered "incorrect"? The authours should not automatically assume poor quality and remove data simply because the values are unusually high, especially if the quality check did not flag them.

l.204: I suggest rephrasing it: Some floats recorded unusually high pCO2 values, as pCO2 values in the Southern Ocean typically range from X to Y (ref). However, considering the "good" quality classification of these data by GDAC standards, and the minimal impact on pCO2 averages (<0.1 uatm) upon their removal, we opted not to exclude them from our analyses." Did you investigate where the high pCO2 data were measured? What regions? Do the authors discuss the increased values later in the manuscript?

L.208: For consistency, I recommend rephrasing it: Increases/decreases in O2 are often accompanied by decreases/increases in pCO2 due to photosynthesis/respirations

l-208-210: Keep your sentences simple. I recommend rephrasing it, e.g.: Although ship-based and float-based O2 values are similar, pCO2 values derived from floats are higher than those derived from ships.

Figure 4: Same comment as for Figure 2. Remove this figure and add error bars to figure 5.

l.215-217 Please remove these (redundant sentences), as the information they convey is already visible in the figure.

l. 218: If the difference is not statistically significant, it may be more appropriate to describe it as "substantial" rather than "significant."

l.223: Points mask axes labels and best-fit lines. Please just add error bars.

l.225: belongs to discussion

l. 232: from "the" atmospheric equilibrium

l.249: I suggest explaining in the method section why the Drake Passage was chosen and consider examining the distribution of the data rather than solely comparing averages.

l.270: "is" reduced

l.271: By removing "a seasonal bias" or "seasonal biases"

l.283: remove "some"

l.289: "substantial" and not "significant"

l.290: Could you clarify why Lueker et al.'s constants were used instead of others?

l. 298: missing blank space

l.318: if you are not accounting for a spatial bias, then you can't attribute the difference to data quality issues

l.325: "2. Seasonal bias in ship-collected data" I would suggest using another word than "bias"

l. 335 & 346: I recommend using "e.g., seasonal bias" instead of "factors 2 & 5". It's only a few more words but significantly improves clarity.

l.352: incomplete sentence

l.354-361: Consider removing as it contains repeated information.

l.385: 4.5 Implications might be more suitable for Introduction and/or Discussion/Conclusions

l.388: Implication (1) does not directly stem from the presented results, so consider revising it accordingly.

l.399: The sentence is misleading. Bushinsky et al 2019 conducted 4 µatm offset experiments/tested whether introducing a 4-µatm offset to the float $p\mathrm{CO_2}$ estimates improves it. However, they found "that the fit of the mapped products to the observations cannot confirm or disprove the existence of a 4 µatm bias in the SOCCOM observations." While Bushinsky did artificially reduce the float pCO2 by 4 uatm, I would add that they did this as a test of sensitivity to a possible bias in float pCO2

l.411: Please add the time periods of the data

l.426. Please add that you are talking about means

l.419 As previously stated, I do not believe this conclusion is supported by the analysis provided. The analysis appears insufficient to justify such a claim.

l.421 You propose that the calculation scheme of pCO2 from float pH requires further refinement and that all sources of uncertainty should be considered, yet you do not provide suggestions for a refinement.

---

## Referee Comment (RC2)

Zhang review

This paper is focused on assessing possible biases in pCO2 values computed from profiling float pH data and estimates of total alkalinity derived from various empirical models fitted to GLODAPv2, shipboard data.This is the third paper from the lead authors on this topic (Wu et al., 2022; 2023). It reiterates some earlier analyses, adds a few new examples, but doesn't add much fundamental new information.

There is no real doubt that there are biases in the profiling float pCO2 estimates, relative to highly calibrated observations made from ships.  The biases, when correctly assessed, generally appear to be within the error limits set in the original Williams et al. paper. That is 2.7% (standard uncertainty) of the pCO2 value.  The critical issue, though, is whether the autonomous observations are useful, despite the biases. The paper doesn't contribute any real understanding in this regard.  It seems more oriented to providing a quite negative perspective. It does this by first reporting uncorrected comparisons, such as those in Figure 3, and stating the bias in float pCO2 is 37 µatm (line 201). Only after that do the authors ultimately apply a set of corrections for age of the ship data, etc., and then, pages later, find the bias from this comparison is 9 µatm (line 332, Figure 9). Nor do they make it easy for the reader to appreciate that the shipboard pCO2 values that they are using as a target also likely have significant biases. The shipboard values are not direct measurements. They are derived from dissolved inorganic carbon and alkalinity with their own, attendant uncertainty.

Perhaps the most robust bias assessment in the manuscript is the direct comparison to SOCAT pCO2 measurements in the Drake Passage, where they find a bias (float minus SOCAT) of 8.3 µatm.  A value of 8.3 µatm is about what we would estimate (5 to 7 µatm with the most recent SOCAT data from the Southern Ocean) and that value is still within the error limits of 11 µatm at near surface pCO2 values reported by Williams et al.  That is nowhere near the values they state on line 201 (37 µatm) and various other places in the manuscript.  The SOCAT comparison is not particularly a new result since values from a similar approach (but more rigorous cross over requirements) are reported in Bushinsky and Cerovecki (2023).  It is also much less comprehensive than the Bushinsky and Cerovecki result, which considered all SOCAT crossover data.

Finally, despite the very negative assessment of the paper, the pCO2 values reported in the paper are proving quite useful.  Examples would be the papers by Prend et al. (2022; doi: 10.1029/2021GB007226), Chen, Haumann et al. (2022; doi: 0.1029/2021GB007156), or a new paper Carranza et al. (provisionally accepted, Extratropical storms induce carbon outgassing over the Southern Ocean).

Another way to look at the value of the float pCO2 estimates is a comparison of float pCO2 with the machine learning products trained on SOCAT data (Johnson et al., presentation at the 2024 Ocean Sciences Meeting). The machine learning products are the values actually used to compute air-sea CO2 flux.  The following figure shows the monthly mean differences in all float surface pCO2 estimates from 70S to 50S (the authors area of focus) with the MPI SOM-FFN values in this region and in each unique month.  During Southern Ocean summer, when SOCAT data are available, the float minus MPI offset is about what one would expect from the float minus SOCAT comparisons. The mean is about 7 µatm. But in winter, the differences increase

dramatically to  as high as 25 µatm.  There are really only two explanations for this.  One is that the float pCO2 bias increases in winter. There is no apparent explanation for that.  The second is that machine learning methods, with little to no data to constrain them in winter, also have a bias. In this case, the machine learning value is underestimating the pCO2 increase observed by floats during winter.

[Figure]

A second example would be the recent paper by Hauck et al. (2023; doi: 10.1029/2023GB007848). In their Figure 5, the air-sea CO2 flux during Southern Ocean summer that is computed from a machine learning product that merges ship and float data is no different than a flux computed with a product based on ship only data.  But in Southern Ocean winter (Hauck Figure 4), the ship only product has a much lower outgassing (pCO2 too low?) in the 50 to 70 S region when compared to the ship plus float product.

I'd argue these results demonstrate the value of float pCO2 estimates.  That is not a message one would get from this paper.

In summary, a significant paper of this type does would do one of three things.  It might identify an unknown bias. It might report a known bias and propose a solution (e.g. the various papers on air-oxygen recalibration of profiling float oxygen sensors). Or it might assess whether the data are useful despite the bias This paper does none of these. The general assessment of bias reported here has been discussed in more than 10 other papers, including 2 by these authors, and is well known. The mean bias of those assessments is generally consistent with the error

estimates in Williams et al. It is also very unclear from this paper what the authors think the bias is; 37 µatm (line 201), 8.3 µatm (line 346), etc.  Finally, it clearly doesn't appreciate that there is significant value in the pCO2 estimates derived from float profiles. These observations lead me to  conclude the paper contributes little to our understanding of pCO2 or air-sea CO2 flux.

Ken Johnson
MBARI

---

## Author Comment (AC1)

**Response to anonymous referee 1:**
(https://doi.org/10.5194/egusphere-2023-3143-RC1)

We thank the reviewer for their feedback and constructive comments on our manuscript. Our responses are in blue below.

**Summary:**

The authors compare the means of pCO2 data measured by ships (1972-2021) with pCO2 data derived from Argo floats (2014-2023) in the Southern Ocean. They find an increase in the float pCO2 values compared to the ship pCO2 and explain the mean difference by seasonality, trends in atmospheric CO2, differences in sampling location, errors in TA, and the choice of carbonate system constants. Consequently, they adjust the pCO2 values by removing the influence of these factors on the mean (e.g., normalizing the data to a reference year). They attribute the remaining difference in mean pCO2 to quality issues. While I appreciate the concept of comparing float pCO2 data with ship pCO2 data, I have major issues with this study. My greatest concerns are as follows:

1. **Content**: The study falls short in making a meaningful contribution to the existing knowledge base. It lacks the presentation of any novel findings. (Higher pCO2 observed in Argo float data than ship data → partially caused by seasonality, different sampling location etc). The conclusion of "bad data quality" appears inadequate given the methodology and is insufficiently discussed).

   Response: we agree with the reviewer that we did not adequately convey the validity of our methods and the novelty of our findings to readers. The reviewer is correct that there have been multiple studies comparing $pCO_2$ estimates from different observational platforms in surface waters. The possibility of a discrepancy in sea surface $pCO_2$ data between float-based and ship-based approaches has been considered and investigated in a number of previous studies (Jin et al., 2024, Wimart-Rousseau et al., 2023, Wu and Qi, 2022, Sutton et al., 2021, Mackay and Watson, 2021, Long et al., 2021, Fay et al., 2018). As suspected by some of these authors, we have indeed identified significant deviations that merit further investigation.

   Our novel finding sheds light and provides greater context on the important and controversial question as to why the air-sea $CO_2$ annual flux calculated from float data is inconsistent with that from other platforms. It remains to be explained why other observations (e.g. from unmanned surface vehicles,

reconstructed pseudo-observations and aircraft-based flux results) are in better agreement with sparse ship observations but differ significantly from float data (Sutton et al., 2021, Mackay and Watson, 2021, Long et al., 2021, Jin et al., 2024). The data collected by these different modalities are broadly consistent, whereas fluxes based on float data suggest significantly lower carbon dioxide uptake. Moreover, sensors on aircraft have not detected the carbon dioxide outgassing at high latitudes in the Southern Ocean predicted by the float data (Long et al., 2021). This study is the first attempt at a novel approach to assessing float data quality and it suggests that the answer to the question is a bias in average float $pCO_2$. The amount of float data we used to check for discrepancies exceeds that used in previous comparison methods. Further justification of our approach and exploration of uncertainty sources are given below and will be added to an amended version of the manuscript (we concur that it needs further discussion).

2. **Methodology:** I fail to understand the rationale behind comparing data from various time periods, seasons, and sampling locations in the first place, particularly when focusing solely on the mean values. In my opinion, this approach is simply not acceptable, as e.g. ocean biogeochemistry undergoes changes over time, leading to higher CO2 levels in more recent float data. While the authors acknowledge this in their later analyses presented in the discussion section, the results in the results section of the study are therefore not comparable. Additionally, the study does not quantify the sources of uncertainty in float pCO2 data, rendering the conclusions regarding data quality issues questionable.

Response: in the following, we describe shortcomings of the main method used previously for comparing $pCO_2$ data from float and ship. We also justify the method applied in this study and explain why it is novel, appropriate and useful.

The main method that has been used prior to this work to assess float data quality is crossover comparisons, i.e. direct comparison of ship and float data when measurements from both are made at the same place and time. While of course valuable, unfortunately there are limitations to this approach. Firstly, ships are only very rarely in the same place as a float at the same time. Therefore, only a very small proportion of the total amount of data can be used in crossover analyses (less than 1%, up to 2023 December). In contrast, our approach compares the totality of float and ship data. Secondly, nearly all the crossover comparisons are made within 3 days of the time that a float

was deployed (Gray et al., 2018, Johnson et al., 2017), because that is the only time when a ship and a float are likely to be coincident. Crossover comparisons made almost exclusively at time of deployment cannot assess lifetime performance of float pH sensors (and thus the $pCO_2$ estimates that are derived) whereas our method can. Some additional analyses (part of future work intended for another manuscript, but shown below) suggest that float age dramatically affects the coherence of float $pCO_2$ data while the oxygen data shows excellent agreement between young and old floats (Figure.1). Although neither overall approach is without shortcomings, our bulk data comparison method is an alternative method of assessing float data quality that is able to assess float sensor performance across whole lifetimes of float deployment. We suggest that it is a valuable complement to crossover analysis.

[Figure]

(A)                                    (B)

Figure (1): Crossover comparison between different floats when they are coincidentally adjacent in time and space. (A) $pCO_2$ comparison between floats having conducted < 50 profiles and floats having conducted > 50 profiles. The best-fit line in red is y=0.53x+184 (r=0.51); (B) The same comparison but for $O_2$ from adjacent floats. The best-fit line in red is y=0.99x+0.78 (r=0.97). Points in both scatterplots are coloured according to the difference in numbers of profiles carried out (as an indication of differences in time since deployment). Two floats are considered adjacent when within 400 km in distance and 7 days in time (Wimart-Rousseau et al., 2023).

We present evidence here that justifies our approach. When two means of measuring the ocean are both measuring correctly then we would expect the large-scale patterns across a basin to agree with each other. We show below that this is true for other parameters measured by floats, although in some

cases only after sampling biases are taken into account. We also show that that discrepancies in $pCO_2$ remain even after sampling biases are corrected for. The reviewer does make a good point, and we agree that it would have been useful to show the comparison for other parameters and we will add this to the manuscript.

[Figure]

Figure (2): Salinity(mean) at different depths. The left panel shows the average float and ship salinity at different depths; the right panel shows the difference between the two (float salinity minus ship salinity).

[Figure]

Figure (3): Nitrate(mean) at different depths. The left panel shows the average float and ship nitrate at different depths; the right panel shows the difference between the two (float nitrate minus ship nitrate).

[Figure]

Figure (4): Temperature(mean) at different depths. The left panel shows the average float and ship temperature at different depths; the right panel shows the difference between the two (float temperature minus ship temperature).

[Figure]

Figure (5): Oxygen(mean) at different depths. The left panel shows the average float and ship oxygen at different depths; the right panel shows the difference between the two (float oxygen minus ship oxygen).

The nitrate and salinity profiles show very good consistency between ship data and float data, while the temperature and oxygen profiles are not well aligned. The misalignment in temperature is explained by a latitudinal gradient in temperature and a sampling bias between ships and floats. More ship data comes from areas further south than does float data (Figure.6).

[Figure]

Figure (6): Proportions of float and ship data from between 50°S and 60°S versus south of 60°S.

The plots below show a comparison between ship temperature and float temperature when the data is separated into 2 regions: between 50°S-60°S and south of 60°S.

[Figure]

Figure (7): Temperature(mean) and difference at different depths in different regions. (A) float and ship data located between 50°S and 60°S; (B) float and ship data south of 60°S region.

The temperature discrepancies are explained by latitudinal effects (Figure.7). This raises the question as to whether a latitudinal effect could also explain the pCO₂ discrepancy. The plots below show the effect of latitude on the pCO₂ discrepancy (Figure.8). The pCO₂ difference in surface waters is 7.0 μ

atm between 50-60°S and 15 μatm south of 60°S. The discrepancy exists in both regions and is in line with the average discrepancy derived in our manuscript. We recognise the necessity of adding a discussion of the effect of latitude on average pCO$_2$ discrepancy to the next version of the manuscript. This will be discussed in addition to the other possible sampling biases already considered (seasonal, spatial and temporal).

[Figure]

Figure (8): pCO$_2$(mean) and difference at different depths in different regions. (A) float data and ship data located in  50°S-60°S region; (B) float data and ship data located in  south of 60°S region.

The solubilities of gases dissolved in seawater are mainly controlled by temperature. Oxygen and carbon dioxide gas concentrations therefore tend to be higher in colder waters. To counteract the effect of a potential sampling bias in temperature (due to a greater proportion of ship data coming from further south where waters are colder; Figure 6) potentially leading to a bias in CO$_2$ and O$_2$ gas concentrations), we calculated the O$_2$ saturation anomaly ($\Delta[O_2]$) ($\Delta[O_2] = \left( \frac{[O_2]_{observed}}{[O_2]_{\text{saturation}}} - 1 \right) \times 100\%$) and the CO$_2$ saturation anomaly ($\Delta[CO_2]$) ($\Delta[CO_2] = \left( \frac{[CO_2]_{observed}}{[CO_2]_{\text{saturation}}} - 1 \right) \times 100\%$) and show the results below.

[Figure]

Figure (9): Oxygen saturation anomaly(mean) at different depths. The left panel shows the average float and ship oxygen saturation anomaly at different depths; the right panel shows the difference between the two (float oxygen saturation anomaly minus ship oxygen saturation anomaly).

[Figure]

Figure (10): $[CO_2]$ saturation anomaly(mean) at different depths. The left panel shows the average float and ship oxygen saturation anomaly at different depths; the right panel shows the difference between the two (float $[CO_2]$ saturation anomaly minus ship $[CO_2]$ saturation anomaly).

Based on the saturation anomaly results (which correct for temperature differences), the float $\Delta[O_2]$ is in rough overall agreement with ship $\Delta[O_2]$ (Figure.9). The float surface $\Delta[CO_2]$ is however approximately 2% higher than ship surface $\Delta[CO_2]$, which converts (at $pCO_2$ of 400 $\mu$atm) to a $pCO_2$

difference of around 8 $\mu$atm. After correcting for various effects and possible biases, the calculated discrepancy in pCO$_2$ is thus close to the results in our manuscript. The large-scale patterns across the Southern Ocean are similar between ship and float data for nitrate, salinity, temperature and oxygen (after removal of latitude/solubility effects) (Figure.2-5,7,9). The fact that the large-scale patterns do not agree for pCO$_2$ even after correcting for sampling biases (Figure.8,10) is therefore a point of interest.

We corrected for accumulation of anthropogenic CO$_2$ over time in surface waters by using the same method as Wu et al. (2019) (their section 2.1), which in turn built on methods described by Takahashi et al. (2009) (their section 2.4). Moving the results of this analysis into the results section, as recommended by the reviewer, will make this clearer and we are happy to do this in the amended version.

The uncertainty in each independent float pCO$_2$ data value does not affect our finding that float pCO$_2$ is systematically high; assuming a normal distribution in individual float uncertainties, as our results are based on a significantly large number of data points, and the standard error of the mean (the standard deviation of mean values) decreases as a function of $\left(1/\sqrt{N}\right)$ where $N$ is the number of data points, then the effect of individual point estimate uncertainties becomes negligible. Williams et al. (2017) estimated the uncertainty of an individual float pCO$_2$ value to be around ±11 $\mu$atm when float pCO$_2$ is 400 $\mu$atm. In the figure below we show the probability density function of average float pCO$_2$ minus ship pCO$_2$ from 1000 Monte Carlo iterations. This figure was generated by the following procedure: (1) assume ship average pCO$_2$ to be 400 $\mu$atm, (2) generate 30,000 independent float pCO$_2$ values, each equal to $400 + G(0,11)$, where $G(\mu,\sigma)$ is a random number from a normal (Gaussian) distribution with mean of $\mu$ and standard deviation of $\sigma$, (3) calculate the average float pCO$_2$ and then the difference between ship and float average values, (4) repeat 1000 times to obtain 1000 differences, (5) plot the frequency distribution of the differences. The effect of uncertainty in each single point of float pCO$_2$ data on the difference in the final float mean is minor (Figure.11). This procedure assumes that errors are random and independent. It does not hold for systematic biases, but that of course is what we are investigating in our study.

[Figure]

Figure (11): Assessment of the impact of uncertainty in individual float $pCO_2$ data on the uncertainty in the overall value of (float $pCO_2$ – ship $pCO_2$), based on Monte Carlo calculations.

We thank the reviewer for raising this point and will add our response to it to our manuscript. We hope that this, together with the other additions, will be seen to have improved the discussion of the merits of this method and additionally will have addressed the reviewer's concerns that our method is not suitable.

3. **Structure:** Result sections 3.1, 3.2 and 3.3 should be merged as the subsections merely contain different plots. The discussion section comprises the presentation of additional analyses, thereby resembling more of a result section.

   Response: we agree with this suggestion. We will restructure the results and discussion sections according to this comment in the next version of the manuscript.

4. **Choice of visualization:** The content in Figure 2, 3, S1 as well as 4, 5, S2 could be merged (remove scatterplots, add error bars to line plots).

   Response: we appreciate this suggestion, and we will take it into account and make this change in the next vision of the manuscript.

5. Authors doubt/question data quality without further arguments (l.200-206). After adjusting the means, they did not go into "float pH data quality issues". I would have appreciated a discussion on why the quality is perceived as poor and how it could be improved etc.

Response: we thank the reviewer for this constructive suggestion. We highlight the quality of float $pCO_2$ data (estimated from pH data) because this is the most likely explanation for the finding in our results that float $pCO_2$ exhibits an overall bias in all our analyses, even after considering (and, where relevant, correcting for) various possible sampling biases. Another reason is that no such bias is seen when comparing $O_2$ data corrected for temperature, whereas a significant bias is seen in $pCO_2$ data corrected both for increasing temperature and increasing atmospheric $CO_2$ (figure 10). We hope that the additions we will make to the manuscript (described above) provide the further arguments the reviewer would like to see.

A float $pCO_2$ bias explains well the large difference between the fluxes calculated by the floats and the fluxes calculated by the other observing platforms. We look forward with anticipation to improved calibration of the float pH data and estimated $pCO_2$, but it is not within the scope of this study to suggest how it should be done. Instead, we present new evidence that the float $pCO_2$ is anomalously high through novel methods, bringing new information to an important field of research. Subsequent work will hopefully ascertain the reasons for this and therefore the solutions. The best process for processing float pH data (and from it float $pCO_2$ values) remains open to discussion; we expect that our findings will eventually contribute to higher accuracy of float $pCO_2$ data.

We appreciate the reviewer's many minor, detailed comments and will attend to these in the revised version of the manuscript.

**References:**

FAY, A. R., LOVENDUSKI, N. S., MCKINLEY, G. A., MUNRO, D. R., SWEENEY, C., GRAY, A. R., LANDSCHÜTZER, P., STEPHENS, B. B., TAKAHASHI, T. & WILLIAMS, N. 2018. Utilizing the Drake Passage Time-series to understand variability and change in subpolar Southern Ocean pCO 2. *Biogeosciences,* 15**,** 3841-3855.

GRAY, A. R., JOHNSON, K. S., BUSHINSKY, S. M., RISER, S. C., RUSSELL, J. L., TALLEY, L. D., WANNINKHOF, R., WILLIAMS, N. L. & SARMIENTO, J. L. 2018. Autonomous biogeochemical floats detect significant carbon dioxide outgassing in the high‐latitude Southern Ocean. *Geophysical Research Letters,* 45**,** 9049-9057.

JIN, Y., KEELING, R. F., STEPHENS, B. B., LONG, M. C., PATRA, P. K., RÖDENBECK, C., MORGAN, E. J., KORT, E. A. & SWEENEY, C. 2024. Improved atmospheric constraints on Southern Ocean CO2 exchange. *Proceedings of the National Academy of Sciences,* 121**,** e2309333121.

JOHNSON, K. S., PLANT, J. N., COLETTI, L. J., JANNASCH, H. W., SAKAMOTO, C. M., RISER, S. C., SWIFT, D. D., WILLIAMS, N. L., BOSS, E. & HAËNTJENS, N. 2017. Biogeochemical sensor performance in the SOCCOM profiling float array. *Journal of Geophysical Research: Oceans,* 122**,** 6416-6436.

LONG, M. C., STEPHENS, B. B., MCKAIN, K., SWEENEY, C., KEELING, R. F., KORT, E. A., MORGAN, E. J., BENT, J. D., CHANDRA, N. & CHEVALLIER, F. 2021. Strong Southern Ocean carbon uptake evident in airborne observations. *Science,* 374**,** 1275-1280.

MACKAY, N. & WATSON, A. 2021. Winter air‐sea CO2 fluxes constructed from summer observations of the polar southern ocean suggest weak outgassing. *Journal of Geophysical Research: Oceans,* 126**,** e2020JC016600.

SUTTON, A. J., WILLIAMS, N. L. & TILBROOK, B. 2021. Constraining Southern Ocean CO2 flux uncertainty using uncrewed surface vehicle observations. *Geophysical Research Letters,* 48**,** e2020GL091748.

TAKAHASHI, T., SUTHERLAND, S. C., WANNINKHOF, R., SWEENEY, C., FEELY, R. A., CHIPMAN, D. W., HALES, B., FRIEDERICH, G., CHAVEZ, F. & SABINE, C. 2009. Climatological mean and decadal change in surface ocean pCO2, and net sea–air CO2 flux over the global oceans. *Deep Sea Research Part II: Topical Studies in Oceanography,* 56**,** 554-577.

WILLIAMS, N., JURANEK, L., FEELY, R., JOHNSON, K., SARMIENTO, J. L., TALLEY, L., DICKSON, A., GRAY, A., WANNINKHOF, R. & RUSSELL, J. 2017. Calculating surface ocean pCO2 from biogeochemical Argo floats equipped with pH: An uncertainty analysis. *Global Biogeochemical Cycles,* 31**,** 591-604.

WIMART-ROUSSEAU, C., STEINHOFF, T., KLEIN, B., BITTIG, H. & KÖRTZINGER, A. 2023. Technical note: Enhancement of float-pH data quality control methods: A study case in the Subpolar Northwestern Atlantic region. *Biogeosciences Discuss.,* 2023**,** 1-26.

WU, Y., HAIN, M. P., HUMPHREYS, M. P., HARTMAN, S. & TYRRELL, T. 2019. What drives the latitudinal gradient in open-ocean surface dissolved inorganic carbon concentration? *Biogeosciences,* 16**,** 2661-2681.

WU, Y. & QI, D. 2022. Inconsistency between ship-and Argo float-based p CO 2 at the intense upwelling region of the Drake Passage, Southern Ocean. *Frontiers in Marine Science,* 9.

---

## Author Comment (AC2)

**Response to Ken Johnson:**
(https://doi.org/10.5194/egusphere-2023-3143-RC2)

We thank the reviewer for their feedback and comments on our manuscript. Here are a few points we would like to respond to regarding the review.

1.  We have not argued that float data is useless and negative; on the contrary, we are well aware of and respect the outstanding contribution of float data to understanding the dynamics of carbon in the Southern Ocean. The coverage of floats fills many observational gaps and increases the possibility of understanding biogeochemical processes at high spatial and temporal precision. Our aim in this manuscript is based on the belief that the float data are very precious and valuable. For this reason, our comprehensive examination of float data accuracy is very useful for a proper understanding of carbon fluxes in the Southern Ocean.

2.  In our opinion, the point that "this bias is already well known" is debatable. A great deal of very recent work makes use of float $pCO_2$ data but simply does not consider or discuss possible biases in it (Chen et al., 2022, Claustre et al., 2020, Djeutchouang et al., 2022, Hauck et al., 2023, Keppler and Landschützer, 2019, Landschützer et al., 2023, Menviel et al., 2023, Mo et al., 2023, Nevison et al., 2020, Prend et al., 2022a, Prend et al., 2022b, Swart et al., 2023, Yang et al., 2024, Huang et al., 2023)

    Even in studies that have considered the possible existence of float bias, there is no agreement on the magnitude of float bias and the distribution of floats with bias. Gray et al. (2018) considered the bias to be 3.6 μatm according to the crossover comparison between float data and SOCAT data. However, we verified that due to spatial and temporal limitations of the cross-comparisons, the float data used for the comparisons were only from the first three days of deployment, and that large amounts of data from later periods were not included in the cross-comparisons. Wu and Qi (2022) took the Drake passage as a case study and found the float-based $pCO_2$ values are overall higher than ship-based values in winter, by 6 to 20 μatm (averaged 14 μatm), which can't be fully explained by the upwelling. This study is limited to the Drake Passage region, rather than a basin-scale comparison across the Southern Ocean.

Bushinsky and Cerovečki (2023) compared the mean $\triangle pCO_2$ of SAMW at the time of formation between float, SOCAT and GLODAP data. They found the float based $\triangle pCO_2$ to be 17–20 µatm higher, of which 6 µatm can be explained by the "possible bias" and the remainder attributed to sampling bias. The data compared in this study include only the time and area of SAMW formation (ACC northern) and do not directly compare data for $pCO_2$ across the Southern Ocean, particularly in the high-latitude ASZ region. It is unconvincing to claim that it is a more comprehensive comparison in terms of float $pCO_2$ data examination.

In summary, there is a large body of research that ignores the bias in float $pCO_2$. The magnitude of float $pCO_2$ bias is still unclear. Whether the $pCO_2$ bias is prevalent in floats throughout the Southern Ocean or only in parts of it has not been determined. The cause and solution of the float $pCO_2$ bias have not yet been determined. We further identify $pCO_2$ discrepancies in the subsurface water measurements, which has not been a consensus from previous works. Therefore, this manuscript of basin-scale comparisons certainly can provide new insight into these questions.

3. The key issue for air-sea $CO_2$ fluxes is whether averaged float estimates of $pCO_2$ are accurate rather than whether individual observations are precise (Bushinsky and Cerovečki, 2023). Uncertainty in the individual float data has little effect on the mean value of the bulk data. Williams et al. (2017) estimated the uncertainty of an individual float $pCO_2$ value to be around ±11 µatm when float $pCO_2$ is 400 µatm; Gregor et al. (2019) estimated the uncertainty of GLODAP $pCO_2$ to be 12 µatm at 400 µatm. In the figure below we show the probability density function of average float $pCO_2$ and ship $pCO_2$ from 1000 Monte Carlo iterations as well as the difference. This figure was generated by the following procedure: (1) assuming float average $pCO_2$ to be 400 µatm and ship average $pCO_2$ to be 390 µatm, (2) generating 30,000 independent float $pCO_2$ values, each equal to $400 + G(0,11)$ and 3,000 independent ship $pCO_2$ values (according to the amount of ship and float data used in the study), each equal to $390 + G(0,12)$, where $G(\mu, \sigma)$ is a random number from a normal (Gaussian) distribution with mean of $\mu$ and standard deviation of $\sigma$, (3) calculate the average float $pCO_2$, ship $pCO_2$ and then the difference between ship and float average values, (4) repeat 1000 times to obtain 1000 differences, (5) plot the frequency distribution of the

differences. The effect of uncertainty in each single point of float or ship pCO₂ data on the difference in the final float mean is minor (Figure.1).

[Figure]

Figure (1): Assessment of the impact of uncertainty in individual float pCO₂ and ship pCO₂data on respective averages and the uncertainty in the overall value of (float pCO₂ – ship pCO₂), based on Monte Carlo calculations.

This result is based on the assumption that errors are random and independent. It does not hold for systematic biases, but that of course is what we are investigating in our study.

**Reference:**

BUSHINSKY, S. M. & CEROVEČKI, I. 2023. Subantarctic Mode Water biogeochemical formation properties and interannual variability. *Agu Advances,* 4**,** e2022AV000722.

CHEN, H., HAUMANN, F. A., TALLEY, L. D., JOHNSON, K. S. & SARMIENTO, J. L. 2022. The deep ocean's carbon exhaust. *Global Biogeochemical Cycles,* 36**,** e2021GB007156.

CLAUSTRE, H., JOHNSON, K. S. & TAKESHITA, Y. 2020. Observing the global ocean with biogeochemical-Argo. *Annual review of marine science,* 12**,** 23-48.

DJEUTCHOUANG, L. M., CHANG, N., GREGOR, L., VICHI, M. & MONTEIRO, P. 2022. The sensitivity of pCO 2 reconstructions to sampling scales across a Southern Ocean sub-domain: a semi-idealized ocean sampling simulation approach. *Biogeosciences,* 19**,** 4171-4195.

GRAY, A. R., JOHNSON, K. S., BUSHINSKY, S. M., RISER, S. C., RUSSELL, J. L., TALLEY, L. D., WANNINKHOF, R., WILLIAMS, N. L. & SARMIENTO, J. L. 2018. Autonomous biogeochemical floats detect significant carbon dioxide outgassing in the high‐latitude Southern Ocean. *Geophysical Research Letters,* 45**,** 9049-9057.

GREGOR, L., LEBEHOT, A. D., KOK, S. & SCHEEL MONTEIRO, P. M. 2019. A comparative assessment of the uncertainties of global surface ocean CO< sub> 2 estimates using a machine-learning ensemble (CSIR-ML6 version 2019a)–have we hit the wall? *Geoscientific Model Development,* 12**,** 5113-5136.

HAUCK, J., NISSEN, C., LANDSCHüTZER, P., RöDENBECK, C., BUSHINSKY, S. & OLSEN, A. 2023. Sparse observations induce large biases in estimates of the global ocean CO2 sink: an ocean model subsampling experiment. *Philosophical Transactions of the Royal Society A,* 381**,** 20220063.

HUANG, Y., FASSBENDER, A. J. & BUSHINSKY, S. M. 2023. Biogenic carbon pool production maintains the Southern Ocean carbon sink. *Proceedings of the National Academy of Sciences,* 120**,** e2217909120.

KEPPLER, L. & LANDSCHüTZER, P. 2019. Regional wind variability modulates the Southern Ocean carbon sink. *Scientific reports,* 9**,** 7384.

LANDSCHüTZER, P., TANHUA, T., BEHNCKE, J. & KEPPLER, L. 2023. Sailing through the southern seas of air–sea CO2 flux uncertainty. *Philosophical Transactions of the Royal Society A,* 381**,** 20220064.

MENVIEL, L. C., SPENCE, P., KISS, A. E., CHAMBERLAIN, M. A., HAYASHIDA, H., ENGLAND, M. H. & WAUGH, D. 2023. Enhanced Southern Ocean CO 2 outgassing as a result of stronger and poleward shifted southern hemispheric westerlies. *Biogeosciences,* 20**,** 4413-4431.

MO, A., PARK, K., PARK, J., HAHM, D., KIM, K., KO, Y. H., IRIARTE, J. L., CHOI, J.-O. & KIM, T.-W. 2023. Assessment of austral autumn air–sea CO2 exchange in the Pacific sector of the Southern Ocean and dominant controlling factors. *Frontiers in Marine Science,* 10**,** 1192959.

NEVISON, C. D., MUNRO, D. R., LOVENDUSKI, N. S., KEELING, R. F., MANIZZA, M., MORGAN, E. J. & RöDENBECK, C. 2020. Southern Annular Mode influence on wintertime ventilation of the Southern Ocean detected in atmospheric O2 and CO2 measurements. *Geophysical Research Letters,* 47**,** e2019GL085667.

PREND, C. J., GRAY, A. R., TALLEY, L. D., GILLE, S. T., HAUMANN, F. A., JOHNSON, K. S., RISER, S. C., ROSSO, I., SAUVé, J. & SARMIENTO, J. L. 2022a. Indo‐Pacific sector dominates Southern Ocean carbon outgassing. *Global Biogeochemical Cycles,* 36**,** e2021GB007226.

PREND, C. J., HUNT, J. M., MAZLOFF, M. R., GILLE, S. T. & TALLEY, L. D. 2022b. Controls on the boundary between thermally and non‐thermally driven pCO2 regimes in the South Pacific. *Geophysical Research Letters,* 49**,** e2021GL095797.

SWART, S., DU PLESSIS, M. D., NICHOLSON, S.-A., MONTEIRO, P. M., DOVE, L. A., THOMALLA, S., THOMPSON, A. F., BIDDLE, L. C., EDHOLM, J. M. & GIDDY, I. 2023. The Southern Ocean mixed layer and its boundary fluxes: fine-scale observational progress and future research priorities. *Philosophical Transactions of the Royal Society A,* 381**,** 20220058.

WILLIAMS, N., JURANEK, L., FEELY, R., JOHNSON, K., SARMIENTO, J. L., TALLEY, L., DICKSON, A., GRAY, A., WANNINKHOF, R. & RUSSELL, J. 2017. Calculating surface ocean pCO2 from biogeochemical Argo floats equipped with pH: An uncertainty analysis. *Global Biogeochemical Cycles,* 31**,** 591-604.

WU, Y. & QI, D. 2022. Inconsistency between ship-and Argo float-based p CO 2 at the intense upwelling region of the Drake Passage, Southern Ocean. *Frontiers in Marine Science,* 9.

YANG, X., WYNN‑EDWARDS, C. A., STRUTTON, P. G. & SHADWICK, E. H. 2024. Drivers of Air‑Sea CO2 Flux in the Subantarctic Zone Revealed by Time Series Observations. *Global Biogeochemical Cycles,* 38**,** e2023GB007766.